# Self-Regulation as a Mediator and Moderator Between School Stress and School Well-Being: A Multilevel Study

**DOI:** 10.3390/ejihpe15120259

**Published:** 2025-12-17

**Authors:** Maja Gajda, Aleksandra Jasińska-Maciążek, Paweł Grygiel, Sylwia Opozda-Suder, Roman Dolata

**Affiliations:** 1Faculty of Education, University of Warsaw, Mokotowska 16/20, 00-561 Warsaw, Poland; mj.gajda2@uw.edu.pl (M.G.); a.jasinska@uw.edu.pl (A.J.-M.); rdolata@uw.edu.pl (R.D.); 2Faculty of Philosophy, Jagiellonian University, Golebia 24, 31-007 Cracow, Poland; sylwia.opozda@uj.edu.pl

**Keywords:** school stress, student well-being, self-regulation, hierarchical linear modeling, primary school

## Abstract

This study examines the relationship between school stress and school well-being, focusing on the mediating and moderating role of self-regulation. This cross-sectional study uses hierarchical linear modeling to assess how two aspects of school stress—perceived school stress at the individual level (students’ subjective appraisal of how stressful specific school demands are) and classroom stressor exposure at the group level (the aggregated frequency of stressful events occurring in each classroom)—are linked to student school well-being. The sample included 702 Polish primary school students (Grades 4, 6, and 8, approx. ages 10–15). Results indicate that while higher perceived school stress is associated with lower well-being, classroom-level stressor exposure also contributes to variations in student well-being. Self-regulation was positively associated with school well-being and partly accounted for the association between perceived stress and well-being. However, no significant moderating effect of self-regulation was found, suggesting that while self-regulation helps explain the link between stress and well-being, it does not necessarily attenuate the association between stress and well-being. These findings highlight the importance of both individual self-regulation skills and structural interventions aimed at reducing classroom stressors to promote student well-being.

## 1. Introduction

In today’s educational landscape, fostering an environment that supports students’ well-being is a growing priority ([9]). The emphasis on student well-being has been driven by concerns over rising academic pressure and mental health struggles among students ([32]). In response, many schools have adopted approaches that promote student engagement, purposeful learning, and social-emotional skills ([18]). Given these challenges and priorities, it is essential to understand how student well-being can be effectively supported within school environments.

School well-being is a multidimensional construct that reflects how students experience and evaluate their life at school. Previous research distinguishes four broad components of school well-being: emotional (positive feelings at school), social (supportive peer and teacher relationships), academic (perceived competence and engagement), and physical or environmental aspects (feeling safe and comfortable at school; [17]; [28]). Rather than representing the absence of distress, school well-being emphasizes positive functioning, such as enjoyment of learning, belonging, motivation, and supportive relationships with peers and teachers ([46]; [15]). School well-being is shaped by students’ overall psychological well-being, as their mental state affects how they experience and engage with the school environment ([37]).

A high level of school well-being has been linked to stronger academic engagement and more adaptive social behavior and, according to some studies, to lower absenteeism ([7]; [33]; [56]). Students with a high level of school well-being are typically more involved and motivated at school, which is reflected in greater participation, higher academic achievement, and stronger peer and teacher relationships.

Student well-being requires special attention in the Polish primary-school context, given rising evidence from national surveys indicating elevated levels of school-related stress and psychological difficulties among children and adolescents. For instance, the 2021/2022 Health Behaviour in School-aged Children (HBSC) survey revealed rising school pressure, particularly among girls, with Poland ranking poorly in adolescent well-being metrics across Europe ([22]). The fact that over 40% of Polish students report frequent nervousness and low mood, often linked to school dissatisfaction and perceived high academic demands, underscores the urgency of targeted research. Similarly, findings from the 2022 Programme for International Student Assessment (PISA) showed that Polish 15-year-olds exhibited lower stress resilience and lower life satisfaction than the OECD average, alongside high levels of worry, nervousness, and exam-related anxiety, with more than half agreeing that they “worry about many things” and “get nervous easily.” Moreover, Polish students reported a comparatively low sense of school belonging—a recognized risk factor for diminished well-being ([25]). By examining how individual self-regulation capacities, perceived school stress and school stressor exposure jointly relate to student well-being, the present study provides actionable evidence for prevention efforts, educational policy, and school-based interventions to support mental health in Polish elementary education.

### 1.1. School Stress

School stress is negatively associated with well-being, but they are not simple opposites. School stress is the experience of unpleasant physical and cognitive reactions triggered by both general and specific challenges associated with the school environment ([26]). General stressors refer to academic pressure, social dynamics, and school policies, while specific stressors can be related to exams, tests, grading, and excessive homework ([65]).

Not all school stress is harmful. In fact, certain levels of stress can foster resilience, strengthen problem-solving skills, and enhance emotional growth ([47]). Mild or moderate stress, when experienced in manageable doses and within a supportive environment, can help individuals learn to navigate life challenges. However, stress effects depend on the intensity, frequency and duration of stress exposure, as well as the availability of recovery opportunities.

The accumulation of stressors students face may be associated with a decline in their well-being. Research shows that high academic stress contributes to psychological distress ([29]) and is a key predictor of student burnout ([14]). Beyond intensifying mental strain, elevated school stress is also linked to diminished positive outcomes, such as lower life satisfaction ([16]) and reduced psychological well-being ([27]).

However, not all students experience school stress in the same way. Students’ responses to stressors depend on how they interpret the situation and the resources available to manage it ([12]). For instance, highly reactive individuals may perceive even minor stressors as overwhelming, increasing their risk of psychological distress and the need for structured support ([5]). More broadly, students’ responses to stressors vary based on their coping mechanisms, personality, and social expectations ([2]; [19]). Those with high neuroticism, or maladaptive coping strategies, such as avoidance or emotional suppression, tend to experience stronger stress reactions. Additionally, older students and girls typically report higher stress, possibly due to greater academic expectations ([32]).

According to the transactional model of stress ([30]), stress arises from the interaction between environmental demands and the individual’s primary and secondary appraisals of those demands. Appraisal determines whether a situation is consciously experienced as stressful. However, research in neurobiology and psychophysiology shows that repeated or persistent exposure to demanding or conflictual environments can produce negative psychological and physiological consequences even when individuals do not report feeling stressed ([34], [36]). These effects occur through mechanisms such as low-intensity or automatic appraisals, cumulative allostatic load, gradual depletion of coping resources, and physiological activation that does not always reach conscious awareness. As a result, repeated exposure to classroom stressors (e.g., peer conflicts or high academic demands) may gradually contribute to increased strain, which in turn can undermine students’ well-being over time, even when they do not consciously appraise these situations as stressful.

Furthermore, stress is not just an individual experience but a shared aspect of the school environment, shaped by social norms and institutional practices ([20]). In some schools, certain stressors can be normalized when adversity is treated as a routine part of student life ([10]), causing students to downplay or dismiss their effects on everyday school functioning. However, even when students do not consciously recognize certain situations as highly stressful on the individual level, these factors can still undermine their sense of school well-being. For instance, conflicts with teachers or classmates can erode students’ sense of safety and belonging, both of which are crucial components of well-being ([57]). Thus, understanding the relationship between school stress and school well-being requires considering both students’ subjective perceptions of how stressful school feels and the frequency of stressors occurring at the classroom level.

### 1.2. Self-Regulation

Students respond to school stressors with varying degrees of self-regulation, which involves managing thoughts and emotions in ways that support goal-directed behavior ([39]). Self-regulation involves processes such as executive functions, inhibitory control, and emotion regulation, all of which work together to support adaptive functioning. Self-regulation is distinct from coping: it reflects students’ general capacity to manage emotions and behavior across situations, whereas coping refers to specific strategies used in response to a particular stressful event (e.g., problem-solving, avoidance, seeking support).

Strong self-regulation is associated with positive outcomes across multiple domains, including academic achievement, social competence, and mental health ([48]). Research consistently indicates that self-regulation is a significant predictor of psychological well-being in adolescents, with interventions targeting self-regulatory techniques showing moderate-to-large effects on mental health outcomes ([60]). Studies using self-regulation measures in school-aged children show that higher self-regulation is associated with greater emotional well-being, fewer internalizing and externalizing difficulties, and more positive relationships with peers and teachers ([13]). Longitudinal work further indicates that self-regulation supports the development of both psychological and school-related well-being as children progress through late primary and early secondary school ([37], [38]). These findings suggest that self-regulation already functions as an important correlate of well-being in primary school.

Self-regulation develops in interaction with environmental factors, including the school setting ([39]). Academic pressure can become a source of chronic stress for students ([63]). When experienced chronically, such pressure may negatively affect self-regulation by influencing brain systems involved in emotional and behavioral control ([35]). More specifically, chronic stress exposure has been linked to structural changes in the hippocampus and medial prefrontal cortex (mPFC), key areas for emotional regulation, decision-making, and executive function ([4]). [55] ([55]) found that students in conventional schools, who were exposed to mild but chronic stress related to testing, grading, and competition, had smaller hippocampal and mPFC volumes compared to students in Montessori schools. This suggests that prolonged exposure to academic stress can be associated with difficulties in self-regulation in some contexts.

Although many studies have examined links between self-regulation, stress, and well-being, the specific role of self-regulation as a mediator or moderator in these associations remains insufficiently understood, especially in primary school populations. Existing research with adolescents and adults suggests that self-regulation may mediate the association between stress and well-being ([51]; [66]) and may also moderate it by buffering or amplifying stress effects ([24]; [52]). However, these studies differ widely in their conceptualizations of well-being and often operationalize it as the absence of distress or psychosomatic symptoms, providing a limited view of positive functioning. From a positive development perspective, well-being encompasses not only low ill-being but also growth, engagement, and belonging ([53]). Research examining how self-regulation relates to these broader dimensions of school well-being is scarce, and even fewer studies assess both mediation and moderation within the same model.

Moreover, multilevel studies that simultaneously consider perceived school stress and exposure to school stressors in primary school students are particularly rare, despite the fact that classroom dynamics are known to play a significant role in children’s daily experiences at school. Very little is known about how these multilevel stress processes operate in students. Clarifying these mechanisms is therefore essential for advancing theoretical understanding and informing interventions tailored to primary school learners.

### 1.3. The Present Study

The present study addresses these gaps by investigating the role of self-regulation in the association between school stress and students’ school well-being in Grades 4, 6, and 8. Specifically, we test two complementary mechanisms: (1) mediation—whether self-regulation (or difficulties in self-regulation) helps explain how perceived school stress and classroom-level exposure to stressors translate into lower school well-being; and (2) moderation—whether stronger self-regulation attenuates the negative association of both perceived school stress and classroom-level stressor exposure with well-being.

In addition, we examine the direct associations of perceived school stress and classroom-level exposure to stressors with students’ school well-being. By simultaneously considering the school as a potential source of stress and as a critical setting for well-being promotion, the study aims to advance transactional stress theory and provide evidence-based insights for school policy and preventive interventions.

While perceived stress (the subjective appraisal of stressors) has been widely studied in relation to well-being, research suggests that classroom-level exposure to stressors may also be associated with students’ lower well-being, even when they do not explicitly perceive certain stressors as stressful ([57]). Perceived school stress reflects students’ subjective appraisal of how stressful the experienced school demands are, whereas exposure to school stressors captures the frequency of stressful events occurring in the classroom environment, irrespective of how stressful they were perceived to be. These constructs are related but not identical: students may be exposed to the same stressors but differ in how strongly they appraise them, and classrooms may differ systematically in the overall level of stressor exposure regardless of individual perceptions. Distinguishing these two aspects of stress allows us to model both the subjective experience of stress at the individual level and the objective or structural conditions of the classroom environment at the group level.

The distinction between reported exposure to stressors (frequency of their occurrence) and their perceived stressfulness therefore reflects two theoretically meaningful aspects of the stress process: the environmental demands that students collectively encounter and the individual appraisals through which those demands are interpreted. Incorporating both levels into a multilevel model allows us to capture how classroom-level stressor exposure may contribute to differences in well-being. Therefore, this study examines the relationship between stress and well-being from two perspectives: perceived school stress and exposure to school stressors. To capture these dynamics, we employ hierarchical linear modeling (HLM) to differentiate between individual-level perceived stress and classroom-level exposure to stressors (see Figure 1).

Based on the study objectives, we formulated several hypotheses focusing on the associations between perceived school stress and classroom exposure to stressors, self-regulation, and school well-being. Given that social dynamics within class groups are important for school well-being ([62]), we first examine the extent to which school well-being varies at the class level. Therefore, we hypothesize that a significant proportion of school well-being variance can be attributed to a class unit (H1).

Moreover, given that school stressors, such as academic pressure, peer conflicts, and teacher expectations, are related to students’ school experiences ([3]), we assume that exposure to school stressors is associated with between-class variability in well-being. Therefore, we hypothesize that greater class-level exposure to stressors is negatively associated with school well-being (H2). On a psychological level, students’ perceived school stress should also be considered, as previous studies indicate that it is a key predictor of well-being ([57]). Therefore, we hypothesize that perceived school stress is negatively associated with school well-being (H3).To clarify the theoretical mechanism underlying our hypotheses, we situate self-regulation within two complementary frameworks: the transactional model of stress and coping ([30]) and the contextual model of self-regulation ([39]). Drawing on the transactional model, we conceptualize stress as a process in which students’ cognitive appraisals of school demands determine the degree to which these demands are experienced as stressful, and these stress experiences, in turn, shape students’ emotional and behavioral responses. Within this framework, appraisals that result in higher levels of perceived stress are expected to be accompanied by more frequent negative emotions and dysregulated reactions, which in turn relate to lower well-being, providing a theoretical basis for our mediation hypothesis.

The contextual model further emphasizes that self-regulation develops through ongoing interactions with environmental stressors and supports and that chronic stress may deplete regulatory resources through cognitive overload and emotional strain. Because of this, students who have stronger regulatory skills are generally better able to manage stress, whereas students with weaker regulatory skills may be more affected by the same stressful experiences—which is the idea behind our moderation hypothesis. Together, these frameworks therefore justify examining self-regulation both as a process through which stress-related experiences may affect well-being and as a capacity that may modify the strength of this association.

When stress is frequent or intense enough to tax emotional and cognitive resources, it may reduce students’ regulatory capacity, which in turn affects their well-being—a pattern consistent with a mediation mechanism. At the same time, self-regulation also reflects the capacity to manage emotional arousal and maintain goal-directed behavior; therefore, students with stronger self-regulation may cope better with the negative effects of stress, whereas those with weaker self-regulation may be more affected, which aligns with a moderation mechanism. These two pathways are theoretically compatible rather than mutually exclusive, making it relevant to examine both within the same model.

Therefore, we formulate three interrelated hypotheses on the role of self-regulation in the relationship between perceived school stress and school well-being. First, in line with previous studies ([37]), we expect self-regulation to be positively correlated with school well-being (H4.1).

Second, we hypothesize that self-regulation moderates the relationship between perceived stress and well-being, potentially acting as a protective factor. Thus, we expect the negative association between perceived school stress and school well-being to be weaker among students with higher self-regulation (H4.2).

Third, self-regulation may mediate the relationship between stress and well-being. Prior research suggests that chronic stress depletes cognitive and emotional resources necessary for self-regulation, which may reduce students’ ability to manage stress effectively and maintain well-being ([51]). Based on these findings, we hypothesize that self-regulation mediates the association between perceived school stress and school well-being, such that higher stress is linked to lower self-regulation, which in turn is associated with lower school well-being (H4.3).

Finally, gender and grade level are associated with school well-being and perceived school stress ([19]; [21]). Girls tend to report higher school stress but also greater school well-being compared to boys ([67]). Additionally, stress typically increases and well-being declines in higher grades due to developmental changes and rising academic pressure ([20]). Therefore, including gender and grade as covariates strengthens our analysis by accounting for individual and contextual factors that may shape students’ school experiences.

In summary, by testing the proposed hypotheses, this study will contribute to the refinement of theoretical models of stress and self-regulation in school settings, filling gaps in existing research and offering valuable implications for educational practice. Our findings may also inform targeted interventions aimed at reducing stress exposure, enhancing students’ ability to self-regulate, and fostering supportive school environments that promote well-being and academic success.

## 2. Methods

### 2.1. Participants

In Poland, education begins with a one-year pre-primary program at age six, followed by eight years of primary school (Grades 1–8). After completing primary school, students take an external examination that plays a key role in secondary school admission. Within this system, students are typically educated in stable classroom groups that remain together across grades, with teachers assigned to specific class divisions. These organizational features, together with the eight-grade structure, position Grades 4, 6, and 8 as key checkpoints in the primary school trajectory, which motivated our focus on these cohorts.

In this context, the present study was conducted in public primary schools located in a medium-sized city in central Poland. Schools were recruited in cooperation with the local education authority, which distributed invitations to all public primary schools in the municipality. Participation was voluntary and based on convenience sampling, as schools self-selected into the study. Within participating schools, all students in the target grades were invited to take part, provided that informed parental consent had been obtained (see Section 2.2 for details). No exclusion criteria were applied other than the inability to complete the online questionnaire during the scheduled data collection session.

The final sample comprised 702 Polish adolescents (49.3% girls) from Grades 4 (45.7%), 6 (35.9%), and 8 (18.4%), typically aged 10–15 years. Participants were drawn from all public primary schools in the city, offering a comprehensive picture of the local student population. In total, 52 classroom groups were included: 21 in Grade 4, 21 in Grade 6, and 10 in Grade 8. Class sizes ranged from 5 to 25 students (M ≈ 13.3).

### 2.2. Procedure

Because the participants were minors, the study followed standard ethical procedures for research with children. Prior to data collection, written informed consent was obtained from parents or legal guardians, who received detailed information about the study aims, procedures, and data protection rules. Only students whose parents provided consent were invited to complete the survey during regular class time.

Before beginning the questionnaire, students were reminded that participation was voluntary, that they could skip any question, and that their responses would remain confidential and analyzed only in anonymized, aggregated form. No personal identifying information was collected beyond the classroom code required for multilevel analyses. All procedures complied with national data protection regulations and the General Data Protection Regulation (GDPR). The study protocol was reviewed and approved by the institutional ethics committee (Research Ethics Board of the Faculty of Education, University of Warsaw, No 2024/2).

Students completed an online questionnaire in computer labs, with interviewers present to provide standardized instructions, ensure confidentiality, and answer procedural questions without influencing responses. The survey took approximately 15 min for higher grades and 35 min for lower grades. Afterward, students were thanked and informed about available support resources (e.g., school psychologist, national youth helpline at 116 111).

### 2.3. Measures

#### 2.3.1. Student Subjective Well-Being Questionnaire

The outcome was measured using the Revised and Extended Polish Version of the Student Subjective Well-being Questionnaire (SSWQ-PL-R; [42]), developed originally by [46] ([46]). It includes 16 items across four subscales: joy of learning (e.g., “I feel happy when I am working and learning at school”), academic efficacy (e.g., “I get good grades in my classes”), teacher–student relationships (e.g., “I feel like teachers at my school care about me”), and peer relationships (e.g., “I have many friends at school”). Each subscale contains four items rated on a 4-point Likert scale (1 = “definitely no” to 4 = “definitely yes”). A higher-order CFA showed good fit (RMSEA = 0.056), and total score reliability was high (α_ord_ = 0.90).

#### 2.3.2. Self-Regulation Scale

This predictor variable was measured using the 12-item Polish adaptation of the Self-Regulation Scale validated in primary school students (sSRS; [50]), and developed originally by [41] ([41]). The scale comprises three subscales: managing anger and frustration (e.g., “I have difficulty controlling my temper”), goalsetting and planning (e.g., “Once I have a goal, I make a plan how to reach it”), and impulse control in goal-directed situations (e.g., “I get very fidgety after a few minutes if I am supposed to sit still”). Items are rated on a 4-point Likert scale (1 = “never true” to 4 = “always true”). A higher-order CFA showed acceptable fit (RMSEA = 0.068), and reliability for the total score was α_ord_ = 0.82.

Because direct measurement of adaptive self-regulatory strategies can be difficult in primary-school children, the instrument primarily captures self-regulatory difficulties (dysregulation), such as problems with attention, emotional control, or behavioral inhibition. The statements for the emotional and behavioral dimensions are scored inversely. Thus, higher scores reflect higher self-regulation.

#### 2.3.3. School Environment Stress Questionnaire

Two predictor variables related to school stress were measured using the School Environment Stress Questionnaire (SESQ; [23]), which was developed and psychometrically validated in samples of primary school students. The SESQ consists of 15 items and three subscales, each with five items: peer relationship stress (“Being teased by other students at school, such as being ridiculed, called names, humiliated, or excluded”), teacher-student relationships stress (“Misunderstandings between you and a teacher”), and academic stress related to learning and assessment (“Taking quizzes and tests”). The SESQ uses a two-step response format: students first indicate whether a given school stressor occurred in their classroom, then rate how stressful it was. Each item is assessed both for its occurrence and associated stress level. The reliability for the total score was α_ord_ = 0.86.

#### School Stressor Exposure

To measure this predictor, we used data from the first step of the SESQ item evaluation. In this step, respondents indicate on a dichotomous scale (0 = “no”, 1 = “yes”) whether a specific stressor occurred in the past few months. The results of the higher-order CFA model confirmed excellent fit (RMSEA = 0.028), and the reliability for the total score was high (α_ord_ = 0.86). Based on student data, an aggregate index of school stressor exposure was created for each class division. This variable was used at the group level in regression models.

#### Perceived School Stress

Data from the second step of the SESQ items assessment process were used to measure this variable. At this stage, respondents who reported experiencing a given stressor rated its intensity on a 5-point Likert scale ranging from 0 (“not stressful at all”) to 4 (“extremely stressful”). For psychometric analysis, unexperienced stressors from the first step of assessment and those rated as “not stressful at all” were recoded to 0, ensuring consistency by equating the absence of a stressor with no perceived stress. A higher-order CFA model for this measure showed excellent fit (RMSEA = 0.038), and the reliability of the total score was α_ord_ = 0.91.

Since a higher-order CFA model provided the best fit to the data for each instrument, theta values for the second-order factor were used in further analyses. These theta values represent the latent second-order factor that captures the common variance of the first-order factors.

#### 2.3.4. Demographic Covariates

Two demographic covariates representing students’ sex and grade level were also included. Sex was coded as 0 = “boy”, 1 = “girl”, and 2 = “other”. School grade, as indicating students’ current educational stage within Poland’s eight-year primary school system, was coded as 1 = “grade 4”, 2 = “grade 6”, and 3 = “grade 8”.

### 2.4. Statistical Analysis

Data were analyzed using hierarchical linear modeling (HLM), which accounts for individual-level variability and dependencies within school classes. This approach was warranted due to significant inter-class correlations in school well-being (ICC > 0.10) and the aggregated index of stressor exposure (ICC = 0.126; [6]). HLM modelling allowed us to control for autocorrelation in the results and obtain more accurate estimates of the fixed effects. Such a model also allows us to distinguish between individual effects (e.g., perceived stress) and contextual effects (stressor exposure). This approach is recommended for analyses of hierarchical educational data because it avoids errors arising from the incorrect assumption of observation independence ([31]).

Because all measurement instruments used in this study demonstrated the best fit under higher-order CFA models, we used second-order theta scores as indicators in subsequent analyses. These scores capture the shared variance across the first-order factors and thus offer a more reliable and parsimonious representation of the underlying constructs than individual items or separate first-order factor scores. Using theta scores also reduces measurement error and multicollinearity among subscales, which is particularly important in multilevel regression and path modeling. For these reasons, second-order latent factor scores were used instead of item-level data or first-order factor scores.

Six sequential models were estimated, five hierarchical regressions and one two-level path model, examining predictors of school well-being at the student (L1) and class (L2) levels. Model 0 (baseline) included no predictors and decomposed variance across levels, supporting H1. Model 1 added control variables: student sex (L1) and class grade (L2). In Model 2, class-level exposure to stressors was introduced, with L1 controls, to test H2. Model 3 added students’ perceived stress (L1) to assess its link with well-being while adjusting for class-level stressor exposure and controls, testing H3. Model 4 included self-regulation (L1) as an additional predictor to test its direct association with well-being (H4.1). Model 5 added the interaction between stress and self-regulation (L1), testing whether self-regulation attenuates the association between perceived stress and well-being (H4.2). Finally, a two-level path model was estimated to examine the mediating role of self-regulation in the stress–well-being link (H4.3), incorporating both direct and indirect effects, with sex (L1), grade (L2), and class-level exposure to stressor as controls.

Variables were pre-standardized, so coefficients were interpreted accordingly. Analyses were conducted in Mplus 8.11 ([40]) using the robust maximum likelihood estimator (MLR), with model fit assessed via AIC and sample-size adjusted BIC. 

## 3. Results

Descriptive statistics for all study variables are provided in Appendix A. The unconditional model (Model 0) indicated significant variance in school well-being at both the student (L1) and class (L2) levels (see Table 1). Although the majority of the variance was observed at the student level (σ^2^ = 0.83, *p* < 0.01), the between-class variance was also statistically significant (τ_00_ = 0.16, *p* < 0.01), yielding an intraclass correlation coefficient (ICC) of 0.16. This finding, in line with H1, suggests that approximately 16% of the total variance in school well-being is attributable to class-level factors, justifying the use of multilevel modeling.

In Model 1, demographic variables were introduced: sex at L1 and grade at L2, as the latter exhibited variance only at the class level. Both were dummy-coded, with males and Grade 4 serving as reference groups. Students in Grades 6 and 8 reported significantly lower levels of well-being compared to those in Grade 4, and students in Grade 8 exhibited significantly lower well-being than those in Grade 6 (Δ = −0.24, *p* = 0.03). The difference in well-being between boys and girls was marginally significant (*β* = 0.16, *p* = 0.05), yet sex was retained as a control variable in subsequent models.

Model 2 incorporated class-level exposure to stressors at L2, revealing a significant negative association with school well-being (*β* = −0.25, *p* < 0.01), thus confirming H2. Introducing this variable into the model decreased the regression coefficients for Grade 6 and Grade 8, suggesting that exposure to stressors was related to grade level (as further confirmed in Model 6, see Figure 2). The intercept variance at L2 was reduced to near zero (τ_00_ = 0.002, *p* = 0.876), indicating that variations in class-level stressors and grade level could explain the majority of between-class differences in well-being.

Model 3 introduced perceived school stress at the student level (L1) to examine its contribution to individual differences in school well-being beyond demographic factors and class-level stress exposure. The results indicated a significant negative association between perceived school stress and well-being, demonstrating that students reporting higher levels of perceived school stress tended to have lower well-being scores (*β* = −0.26, *p* < 0.01). This finding confirms H3. Moreover, this model revealed a significant effect of sex, indicating that, when controlling for exposure to stressors and perceived stress, girls exhibited higher levels of school well-being than boys (*β* = 0.26, *p* < 0.01).

In Model 4, self-regulation was added as a fixed effect at the student level (L1) to assess whether it is independently associated with school well-being and whether it altered the previously observed relationship between stress and well-being. The findings, consistent with H4.1, revealed a moderately strong positive association between self-regulation and school well-being, such that students with higher self-regulation tended to report higher well-being (*β* = 0.37, *p* < 0.01). Notably, the inclusion of self-regulation in the model resulted in a substantial reduction in the regression coefficient for perceived stress, suggesting that self-regulation partially mediates the negative impact of stress on well-being.

Model 5 tested an interaction term (stress × self-regulation) to examine whether self-regulation moderates the relationship between stress and well-being. However, no significant moderating effect was detected, and fit indices (AIC, BIC) indicated no improvement over Model 4, suggesting that self-regulation does not function as a moderator in this context, providing no empirical support for H4.2, which was therefore rejected.

Model 6 (see Figure 2) extended the previous analyses by estimating a two-level path model that explicitly tested whether self-regulation mediates the relationship between school stress and well-being, controlling for significant predictors at both student (L1) and class (L2) levels. The results revealed a direct negative effect of perceived stress on well-being (*β* = −0.10, *p* = 0.05) and a notable indirect effect (β_indirect_ = −0.17, *p* < 0.01) via self-regulation, indicating partial mediation. Specifically, high levels of stress were associated with lower self-regulation (*β* = −0.47, *p* = 0.04), which in turn was strongly and positively associated with school well-being (*β* = 0.37, *p* = 0.04). These findings, in line with H4.3, suggest that self-regulation serves a protective function within the mediation process, although it does not entirely eliminate the adverse effects of stress. The statistically significant direct effect further implies that stress continues to exert a unique negative effect on well-being, even after accounting for self-regulation. At the class level, exposure to school stressors remained an important predictor of between-class differences in well-being, underscoring the interplay of individual and contextual factors in shaping students’ experiences.

## 4. Discussion

This study aimed to examine the complex relationship between school stress and school well-being among primary school students, with a particular focus on the role of self-regulation as both a mediator and a moderator. By employing a multilevel analytical approach, we sought to distinguish between individual-level perceived stress and classroom-level exposure to stressors, thus accounting for both personal and environmental factors related to well-being.

Our results confirm that school well-being varies significantly by class (H1), which is consistent with the broader phenomenon of differential class environments ([44]). Supporting H2, we also found that students in classrooms with greater exposure to stressors (L2) reported lower average levels of school well-being, suggesting that both perceived school stress and well-being have a substantial systemic component ([57]). At the individual level, our findings support H3, showing that students who perceive stressors as more intense tend to experience lower well-being. This aligns with existing research linking academic pressure, peer conflict, and external expectations to reduced school well-being ([8]). Together, these findings highlight the importance of addressing both the structural and subjective dimensions of school stress. Promoting well-being requires reducing harmful conditions and supporting adaptive coping.

Regarding H4.1, self-regulation was positively associated with school well-being, consistent with previous findings on its role in promoting emotional stability, social competence, and academic engagement ([38]). However, H4.2, which posited that self-regulation moderates the relationship between stress and well-being, was not supported. The interaction effect was not statistically significant, suggesting that self-regulation did not attenuate the association between perceived stress and well-being in this sample. In this study, self-regulation was assessed as a resource and a general tendency to experience challenges in managing emotions, attention, and behavior, rather than as the use of specific coping strategies (e.g., reappraisal or problem-solving). While this trait-based approach was appropriate for our younger sample (ages 10–15), it may not fully capture the situational coping processes involved in mitigating acute stress. Thus, the absence of a moderation effect may reflect limitations of the construct measured rather than the absence of a buffering mechanism altogether.

Alternatively, the extent to which self-regulatory resources shape the stress–well-being association may depend on contextual factors, such as the type or intensity of stressors and the availability of external support ([39]). Supportive teachers, positive peer relationships, and a nurturing school climate with clear expectations and consistent rule enforcement are known to promote both self-regulation and well-being ([1]; [45]). In the absence of such support, even students who are typically well-regulated may struggle to maintain well-being in high-stress settings. This underscores the importance of structured and sustained environmental support to reinforce students’ regulatory skills, rather than relying solely on their individual self-regulation capacity ([39]).

Our study suggests that self-regulation mediates the relationship between perceived stress and well-being (H4.3). Higher stress was associated with reduced self-regulatory capacity, which in turn was associated with lower well-being. This aligns with neurocognitive research indicating that stress can impair executive functions, increase impulsivity, disinhibition, and emotion dysregulation ([43]). From an educational perspective, these findings suggest that interventions should not only focus on strengthening self-regulation skills but also target stressors that chronically deplete students’ self-regulatory resources.

At the individual level, the observed mediation pattern is compatible with the transactional model of stress ([30]), which describes stress as arising from the appraisal of environmental demands relative to personal resources. In our findings, students who appraised school demands as more stressful also tended to report lower self-regulation, aligning with the notion that heightened stress experiences may coincide with reduced regulatory capacity. Lower self-regulation, in turn, was associated with lower school well-being. This suggests that appraisal-based stress is linked to well-being partly through students’ self-regulatory resources. These findings are generally consistent with prior research conducted with primary school students, although direct comparisons remain limited. Studies using similar age groups have shown that higher self-regulation is associated with greater emotional well-being and more positive school functioning ([49]; [59]). Furthermore, studies examining school stress in upper primary grades have documented that students who report more intense or frequent stressors also tend to report lower school satisfaction and belonging ([17]), broadly supporting the pattern identified in our results. Although few studies have tested mediation or moderation models in this age group, the direction and strength of the associations observed in the present study are consistent with these earlier findings.

As [39] ([39]) suggest, self-regulation skills should be explicitly taught and practiced, much like literacy skills, to ensure that all students develop effective strategies. Evidence from school-based programs shows that specific self-regulation strategies, such as relaxation and cognitive restructuring, often delivered through mindfulness, can reduce psychological stress in students ([60]; [61])**.** Moreover, studies assessing mindfulness interventions have found that self-regulation mediates their positive effects on well-being ([11]; [54]). Together, these findings underscore the central role of self-regulation as both a predictor of well-being and a key mechanism underlying successful stress-reduction programs.

Our findings suggest that students may report low school stress yet still experience reduced well-being when exposed to persistent environmental stressors. Stressors such as peer teasing, teacher conflict, or harsh disciplinary practices can erode trust and belonging, even when students do not consciously experience them as highly stressful. The accumulation of such stressors can create an unfavorable school climate, weakening respect, connectedness, and engagement which are key elements of student well-being. Consequently, improving school well-being requires not only supporting individual stress management but also systematically reducing the presence of certain stressors within the school environment.

It is important to recognize that some degree of stress is inherent to schooling due to its systemic functions. For example, the pervasive nature of grading (in our survey, 91% of students reported being stressed by classwork) and the social dynamics of peer group positioning are inherent aspects of the school environment that cannot be easily eliminated. Importantly, not all school stress is harmful, and certain stress levels can foster resilience, strengthen problem-solving skills, and enhance emotional growth ([47]). However, whether stress has a beneficial or harmful effect depends on its intensity, duration, and the availability of support and recovery opportunities. Cumulative stressors, such as frequent teacher-student conflicts, bullying, or social exclusion, can escalate into toxic stress if they remain unaddressed. Also, individuals with high reactivity to stressors may struggle to cope even with lower-intensity stressors, making them more vulnerable to their negative effects ([5]).

## 5. Limitations and Future Research Directions

Despite its contributions, this study has several limitations. First, our measure of self-regulation primarily assessed difficulties in managing emotions and behavior rather than specific regulatory strategies. Future research should examine whether particular coping mechanisms, such as cognitive reappraisal, problem-solving, or attentional control, moderate the stress-well-being relationship differently.

Additionally, our analysis provides a broad indicator of students’ regulatory capacities but does not capture potential imbalances across cognitive, emotional, and behavioral domains. Future studies could benefit from multidimensional assessments of self-regulation to explore whether specific regulatory strengths or deficits have a greater impact on stress and well-being.

Second, our findings are based on correlational data, so causal conclusions cannot be drawn. While the results suggest that higher stress levels are associated with lower self-regulation, which in turn is linked to reduced well-being, it is also possible that students with weaker self-regulation perceive stressors as more overwhelming, creating a reciprocal relationship. Longitudinal or experimental studies are needed to clarify these dynamics.

Third, although we controlled for gender and grade level, other contextual factors such as teacher support, school climate, and family background may also play an important role in shaping students’ perceived school stress and well-being. Future research should adopt a multilevel approach that includes broader school-wide and policy-level variables to provide a more comprehensive understanding of student well-being.

It should be noted that the present study did not include analyses of measurement invariance across gender or grade levels. Because our aim was not to compare latent means between groups but to estimate within- and between-class associations, establishing cross-group invariance was not required for the primary research questions. Nevertheless, future research examining developmental or gender differences in school stress, self-regulation, or well-being should formally test measurement invariance to ensure the comparability of constructs across groups.

## 6. Implications

A key implication of our findings is that improving school well-being requires systemic changes, not just individual coping strategies. While training in self-regulation (e.g., mindfulness or cognitive restructuring) helps students manage stress ([60]), it may be insufficient if high levels of exposure to school stressors persist. Given the impact of classroom-level stressors, interventions should also reduce systemic stress by fostering supportive environments, easing academic pressure, and encouraging positive peer and teacher interactions ([1]).

These recommendations are further supported by research consistently showing that authoritative school environments, characterized by high support and clear structure, promote students’ emotional well-being and social adjustment ([64]). Students in such environments report lower levels of stress and fewer psychosomatic symptoms, suggesting that clear expectations, positive teacher-student relationships, and access to support serve as protective factors.

Moreover, given evidence that high-stakes testing and performance-driven school cultures contribute to elevated stress and reduced well-being ([20]), educational policy should seek a better balance between academic demands and emotional support. Schools that emphasize mastery experiences over performance orientation tend to produce more favorable psychological outcomes for students, suggesting that revisions to assessment and grading policies may play a key role in promoting school well-being ([58]).

## 7. Conclusions

This study examined how perceived school stress and classroom-level exposure to stressors relate to school well-being in primary school students and whether self-regulation functions as a mediator or moderator in these associations. At the individual level, higher perceived stress was associated with lower self-regulation and lower school well-being, and self-regulation partly accounted for the association between stress and well-being. The interaction term was not significant, indicating that self-regulation did not attenuate the stress–well-being link in this age group. At the classroom level, students in classes with more frequent stressors reported lower well-being on average, even after accounting for perceived stress levels.

These findings highlight the value of examining both individual stress appraisals and classroom stressor exposure when studying well-being in primary school. The results also support a view of self-regulation as a regulatory resource associated with well-being rather than a factor that changes (moderates) the strength of stress–well-being associations. Integrating these processes within a multilevel framework provides a more comprehensive understanding of how stress experiences operate across levels of the school environment.

Although the study was not designed to test interventions, the findings underscore the importance of supporting children’s self-regulatory capacities and reducing classroom stressor exposure as part of broader strategies to promote school well-being. Any recommendations in this area should remain tentative and grounded in the correlational nature of the findings.

By integrating individual and classroom-level indicators of stress, self-regulation, and school well-being, this study contributes novel evidence on the multilevel processes shaping students’ experiences in upper primary grades. Future research would benefit from longitudinal designs or experimental approaches that can test causal pathways and examine how changes in stress, regulatory resources, and classroom conditions unfold over time.

## Figures and Tables

**Figure 1 ejihpe-15-00259-f001:**
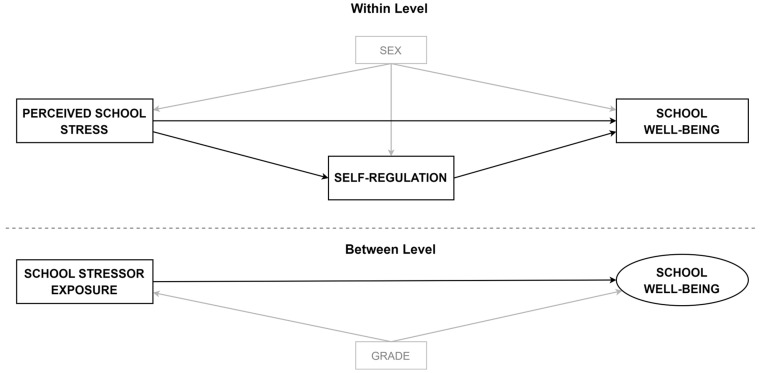
Conceptual model of the mediating and moderating effects of self-regulation on the relationship between school stress and school well-being.

**Figure 2 ejihpe-15-00259-f002:**
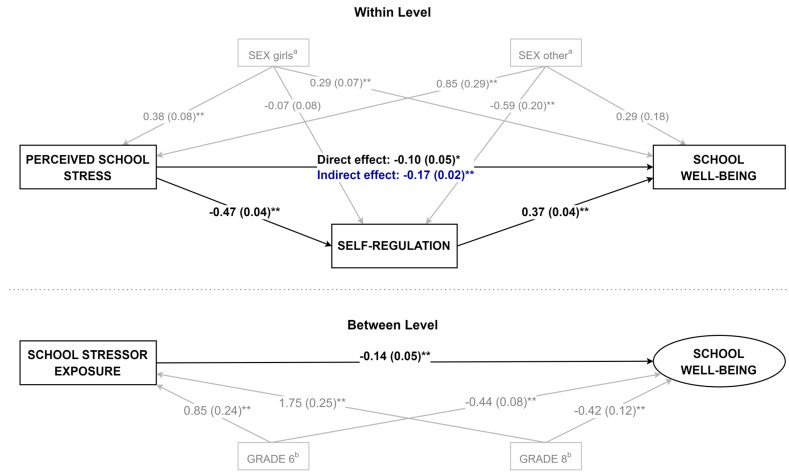
Path model of the mediating and moderating effects of self-regulation on school stress and well-being at within- and between-class levels. Note. Unstandardized coefficients are presented; ^a^ indicates sex-coded predictors (reference category: boys); ^b^ indicates grade-coded predictors (reference category: Grade 4); * *p* < 0.05, ** *p* < 0.01.

**Table 1 ejihpe-15-00259-t001:** Multi-level analysis of the relationship between school stress, self-regulation, and well-being.

Parameters		Model 0		Model 1		Model 2		Model 3		Model 4		Model 5	
		B	(SE)	B	(SE)	B	(SE)	B	(SE)	B	(SE)	B	(SE)
Fixed effects													
Within level (L1)													
	Intercept	−0.014	(0.066)	0.321 **	(0.083)	0.147	(0.082)	0.092	(0.079)	0.083	(0.079)	0.082	(0.079)
	Sex girls			0.161	(0.083)	0.170 *	(0.082)	0.264 **	(0.075)	0.290 **	(0.069)	0.289 **	(0.067)
	Sex other			−0.152	(0.250)	−0.141	(0.247)	0.041	(0.200)	0.288	(0.177)	0.283	(0.167)
	Perceived stress							−0.261 **	(0.044)	−0.098 *	(0.049)	−0.098 *	(0.048)
	Self-regulation									0.367 **	(0.040)	0.367 **	(0.039)
	Perceived stress × Self-regulation											−0.004	(0.032)
Between level (L2)													
	Grade 6			−0.651 **	(0.095)	−0.434 **	(0.080)	−0.421 **	(0.083)	−0.444 **	(0.085)	−0.444 **	(0.083)
	Grade 8			−0.889 **	(0.105)	−0.434 **	(0.118)	−0.428 *	(0.117)	−0.423 **	(0.121)	−0.425 **	(0.125)
	Stressor exposure					−0.255 **	(0.049)	−0.175 **	(0.052)	−0.145 **	(0.049)	−0.144 **	(0.050)
Random effects													
	Residual (L1) variance	0.829 **	(0.044)	0.820 **	(0.042)	0.813 **	(0.042)	0.756 **	(0.039)	0.648 **	(0.036)	0.648 **	(0.036)
	Intercept (L2) variance	0.158 **	(0.036)	0.026	(0.016)	0.002	(0.015)	0.000	(0.011)	0.009	(0.010)	0.009	(0.010)
Additional information													
	Number of class	52		52		52		52		52		52	
	Number of students	692		692		692		692		692		692	
	Log-likelihood	−949.080		−922.404		−911.108		−885.334		−836.090		−836.077	
	Scaling correction factor (MLR)	0.8577		0.9891		0.9521		1.0208		1.0146		1.0491	
	AIC	1904.161		1858.809		1838.217		1788.669		1692.180		1694.154	
	Sample-Size Adjusted BIC	1908.254		1868.360		1849.132		1800.948		1705.824		1709.163	
	Number of est. parameters	3		4		8		9		10		11	
	Pseudo R^2^ (L1)			0.009	(0.008)	0.010 **	(0.008)	0.088 **	(0.026)	0.217 **	(0.029)	0.217 **	(0.029)
	Pseudo R^2^ (L2)			0.837 **	(0.093)	0.986 **	(0.086)	0.997 **	(0.095)	0.918 **	(0.082)	0.917 *	(0.080)

Note. B = beta; SE = standard error; L1 = Level 1; L2 = Level 2; Log-likelihood reflects the maximized log of the likelihood function for each model; the scaling correction factor (MLR) adjusts the log-likelihood and chi-square statistics for non-normality under the robust MLR estimator; The pseudo-R^2^ values obtained using MLR may differ from those calculated using the traditional ML method. * indicates *p* < 0.05; ** indicates *p* < 0.01.

## Data Availability

The data that support the findings of this study are openly available in RepOD at: https://repod.icm.edu.pl/dataset.xhtml?persistentId=doi:10.18150/ENODB5 (accessed on 12 December 2024).

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
