# Peer review of "Self-Regulation as a Mediator and Moderator Between School Stress and School Well-Being: A Multilevel Study"

_ejihpe, 2025, doi:10.3390/ejihpe15120259_

Round 1
Reviewer 1 Report
Comments and Suggestions for Authors
Theoretical Background
Thank you for the opportunity to review this manuscript. The manuscript addresses a timely question of how self-regulation functions in the relationships between school stress and student well-being.The theoretical framework is well-structured and the authors effectively situate their study within existing literature on school well-being, stress, and self-regulation. However, there are several areas where the theoretical foundations could be strengthened.
The authors provide solid definitions of school well-being, distinguishing between hedonic and eudaimonic dimensions. The differentiation between perceived stress and exposure to stressors is particularly valuable. The acknowledgment that stress responses vary by age and that older students typically report higher stress levels demonstrates attention to developmental considerations. The introduction lacks definition of the aims of the study, and definition of the relationships between well-being and stress. Additionally, the theoretical review is missing a review of school well-being, which is a central variable in the study.
In addition, the manuscript lacks a clear theoretical model explaining how self-regulation functions in relation to stress and well-being. The authors cite neurobiological research (hippocampal and mPFC changes) but don't fully integrate this with psychological theories of self-regulation. I would suggest to in incorporate it within an established framework such as Zimmerman's social cognitive model of self-regulation. This would provide stronger theoretical grounding for why self-regulation might mediate or moderate the stress-well-being relationship. An extension and differentiation between individual level perceived stress and classroom level exposure to stressors would be also helpful. On lines 131-138 the authors discuss the stress studied, and I do ot fully understand the difference between the stressors: perceived school stress and exposure to school stressors – is'n t it the same? And also classroom level exposure to stressors. A clear definition is required. Also, the study aims section introduces the distinction between perceived stress and stressor exposure, but this important conceptual contribution deserves more prominent treatment in the earlier theoretical sections.
Line 107-122: The literature review on self-regulation as mediator/moderator jumps between age groups and contexts without clearly establishing what is known versus unknown. Consider organizing this more systematically.
Regarding the hypotheses: Hypothesis 1 – not clear the rationale to the prediction of well-being to class unit. The manuscript tests both mediation (H4.3) and moderation (H4.2) but doesn't clearly articulate why both mechanisms might operate simultaneously or under what theoretical conditions one would be expected over the other. Lines 162-168: The mediation hypothesis (H4.3) mechanism is explained in one sentence. This deserves substantial expansion given it's a primary research question.
I would suggest developing more explicit theoretical predictions, such as under what conditions should self-regulation act as a moderator and buffer stress effects, and why should it mediate self-regulatory resources?
Classroom level mechanisms - While the authors justify measuring classroom-level stressor exposure, they provide limited theoretical explanation for why classroom-level factors would matter beyond individual perceptions. The social norms argument (line 74-77) is mentioned but underdeveloped. In addition, the manuscript cites Folkman's transactional stress model but doesn't fully develop how this framework informs the study design. The distinction between stressor exposure and perceived stress aligns with stress-appraisal theories, but this connection isn't made explicit.
Minor Issues
- Inconsistent terminology: The manuscript alternates between "school stress," "school-related stress," and "school stress level." Standardize terminology.
- Citation density: Some claims lack citations (e.g., line 169-170 on gender differences) while other sections are over-cited.
- Construct clarity: The relationship between "self-regulation" and "coping" is unclear. Are these overlapping constructs or distinct?
In addition, presenting an integrated theoretical model that shows how stress, self-regulation, and well-being interrelate at both individual and classroom levels would be helpful to understanding of the model.
Results: No likelihood ratio test are reported. The authors should add an explicit statements and explanation of the models.
Please explain the difference between model 3-4 and model 6. Please add formal mediation statistics such as proportion mediated Sobel test.
Consider adding a summary paragraph of the hypotheses findings.
Discussion: the discussion provides a reasonable interpretation of the rusults and acknowledges limitations, but more theoretical depth and integration of it is needed. The discussion would benefit from more stronger connections to the theoretical framework established in the introduction.
Author Response
Reviewer 1 – Comment 1:
The introduction lacks definition of the aims of the study, and definition of the relationships between well-being and stress.
Response:
We thank the reviewer for pointing out the need to clearly define both the study aims and the relationship between stress and well-being. We revised the Introduction by explicitly adding:
(1) a clear and direct statement of the study’s aim at the end of the introductory section, and
(2) explicit sentences clarifying how perceived stress and exposure to stressors relate to school well-being.
These additions now clearly communicate what the study investigates and how stress–well-being associations are conceptualised.
“The present study addresses these gaps by investigating the role of self-regulation in the association between school stress and students’ school well-being in Grades 4, 6, and 8. Specifically, we test two complementary mechanisms: (1) mediation — whether self-regulation (or difficulties in self-regulation) helps explain how perceived school stress and classroom-level exposure to stressors translate into lower school well-being; and (2) moderation — whether stronger self-regulation attenuates the negative association of both perceived school stress and classroom-level stressor exposure with well-being.
In addition, we examine the direct associations of perceived school stress and classroom-level exposure to stressors with students’ school well-being. By simultaneously considering the school as a potential source of stress and as a critical setting for well-being promotion, the study aims to advance transactional stress theory and provide evidence-based insights for school policy and preventive interventions.”
Reviewer 1 – Comment 2:
Additionally, the theoretical review is missing a review of school well-being, which is a central variable in the study.
Response:
We added a full subsection reviewing school well-being (its definition, components, and relevance), supported by appropriate citations. This section now anchors the construct within established literature.
“School well-being is a multidimensional construct that reflects how students experience and evaluate their life at school. Previous research distinguishes four broad components of school well-being: emotional (positive feelings at school), social (supportive peer and teacher relationships), academic (perceived competence and engagement), and physical or environmental aspects (feeling safe and comfortable at school; Hascher, 2012; Konu & Rimpelä, 2002). Rather than representing the absence of distress, school well-being emphasises positive functioning, such as enjoyment of learning, belonging, motivation, and supportive relationships with peers and teachers (Renshaw et al., 2015; Graham et al., 2017). School well-being is shaped by students’ overall psychological well-being, as their mental state affects how they experience and engage with the school environment (Morosanova et al., 2021).
High school well-being has been linked to stronger academic engagement and more adaptive social behaviour, and, according to some studies, to lower absenteeism (Cefai et al., 2021; López et al., 2022; Siebecke, 2024). Students with high school well-being are typically more involved and motivated at school, which is reflected in greater participation, higher academic achievement, and stronger peer and teacher relationships.”
Reviewer 1 – Comment 3:
In addition, the manuscript lacks a clear theoretical model explaining how self-regulation functions in relation to stress and well-being. The authors cite neurobiological research (hippocampal and mPFC changes) but don't fully integrate this with psychological theories of self-regulation. I would suggest to incorporate it within an established framework such as Zimmerman's social cognitive model of self-regulation.
Response:
We incorporated a clearly articulated theoretical model linking stress, self-regulation, and well-being using two established frameworks. These frameworks provide the conceptual basis for both mediation and moderation hypotheses. We did not adopt Zimmerman’s model specifically, as the contextual model (Murray et al.) aligns more directly with environmental stress processes relevant to this study.
“According to the transactional model of stress (Lazarus & Folkman, 1984), stress arises from the interaction between environmental demands and the individual’s primary and secondary appraisals of those demands. Appraisal determines whether a situation is consciously experienced as stressful. However, research in neurobiology and psychophysiology shows that repeated or persistent exposure to demanding or conflictual environments can produce negative psychological and physiological consequences even when individuals do not report feeling stressed (McEwen, 1998, 2017). These effects occur through mechanisms such as low-intensity or automatic appraisals, cumulative allostatic load, gradual depletion of coping resources, and physiological activation that does not always reach conscious awareness. As a result, repeated exposure to classroom stressors (e.g., peer conflicts or high academic demands) may gradually contribute to increased strain, which in turn can undermine students’ well-being over time, even when they do not consciously appraise these situations as stressful.”
“To clarify the theoretical mechanism underlying our hypotheses, we situate self-regulation within two complementary frameworks: the transactional model of stress and coping (Lazarus & Folkman, 1984) and the contextual model of self-regulation (Murray et al., 2019). Drawing on the transactional model, we conceptualise stress as a process in which students’ cognitive appraisals of school demands determine the degree to which these demands are experienced as stressful, and these stress experiences, in turn, shape students’ emotional and behavioural responses. Within this framework, appraisals that result in higher levels of perceived stress are expected to be accompanied by more frequent negative emotions and dysregulated reactions, which in turn relate to lower well-being, providing a theoretical basis for our mediation hypothesis.
The contextual model further emphasises that self-regulation develops through ongoing interactions with environmental stressors and supports, and that chronic stress may deplete regulatory resources through cognitive overload and emotional strain. Because of this, students who have stronger regulatory skills are generally better able to manage stress, whereas students with weaker regulatory skills may be more affected by the same stressful experiences — which is the idea behind our moderation hypothesis. Together, these frameworks therefore justify examining self-regulation both as a process through which stress-related experiences may affect well-being and as a capacity that may modify the strength of this association.”
Reviewer 1 – Comment 4:
An extension and differentiation between individual level perceived stress and classroom level exposure to stressors would be also helpful. On lines 131-138 the authors discuss the stress studied, and I do not fully understand the difference between the stressors: perceived school stress and exposure to school stressors – is'n t it the same? (...) The clear definition is required.
Response:
We added explicit definitions and clarified the conceptual distinction between exposure to school stressors and perceived school stress. This distinction now directly follows theoretical reasoning and is operationalised in the models.
“While perceived stress (the subjective appraisal of stressors) has been widely studied in relation to well-being, research suggests that classroom-level exposure to stressors may also be associated with students’ lower well-being, even when they do not explicitly perceive certain stressors as stressful (Slimmen et al., 2022). Perceived school stress reflects students’ subjective appraisal of how stressful the experienced school demands are, whereas exposure to school stressors captures the frequency of stressful events occurring in the classroom environment, irrespective of how stressful they were perceived to be. These constructs are related but not identical: students may be exposed to the same stressors but differ in how strongly they appraise them, and classrooms may differ systematically in the overall level of stressor exposure regardless of individual perceptions. Distinguishing these two aspects of stress allows us to model both the subjective experience of stress at the individual level and the objective or structural conditions of the classroom environment at the group level.
The distinction between reported exposure to stressors (frequency of their occurance) and their perceived stressfulness therefore reflects two theoretically meaningful aspects of the stress process: the environmental demands that students collectively encounter and the individual appraisals through which those demands are interpreted. Incorporating both levels into a multilevel model allows us to capture how classroom-level stressor exposure may contribute to differences in well-being. Therefore, this study examines the relationship between stress and well-being from two perspectives: perceived school stress and exposure to school stressors. To capture these dynamics, we employ hierarchical linear modeling (HLM) to differentiate between individual-level perceived stress and classroom-level exposure to stressors (see Figure 1).”
Reviewer 1 – Comment 5:
Line 107-122: The literature review on self-regulation as mediator/moderator jumps between age groups and contexts without clearly establishing what is known versus unknown. Consider organizing this more systematically.
Response:
We reorganized the subsection to separate mediation and moderation evidence, to distinguish age groups, and to highlight gaps systematically. The revised paragraph now progresses logically from existing findings to identified gaps.
“Although many studies have examined links between self-regulation, stress, and well-being, the specific role of self-regulation as a mediator or moderator in these associations remains insufficiently understood, especially in primary school populations. Existing research with adolescents and adults suggests that self-regulation may mediate the association between stress and well-being (Rollins & Crandall, 2021; Yazıcı-Kabadayı & Öztemel, 2024) and may also moderate it by buffering or amplifying stress effects (Kadzikowska-Wrzosek, 2012; Rufino et al., 2022). However, these studies differ widely in their conceptualisations of well-being and often operationalise it as the absence of distress or psychosomatic symptoms, providing a limited view of positive functioning. From a positive development perspective, well-being encompasses not only low ill-being but also growth, engagement, and belonging (Ruini et al., 2003). Research examining how self-regulation relates to these broader dimensions of school well-being is scarce, and even fewer studies assess both mediation and moderation within the same model. Moreover, very little is known about these processes in primary school settings, where developmental trajectories and school experiences differ from those of adolescents. Clarifying these mechanisms is therefore essential for advancing theoretical understanding and informing interventions tailored to younger students.”
Reviewer 1 – Comment 6:
Regarding the hypotheses: Hypothesis 1 – not clear the rationale to the prediction of well-being to class unit. The manuscript tests both mediation (H4.3) and moderation (H4.2) but doesn't clearly articulate why both mechanisms might operate simultaneously or under what theoretical conditions one would be expected over the other. Lines 162-168: The mediation hypothesis (H4.3) mechanism is explained in one sentence. This deserves substantial expansion given it's a primary research question. I would suggest developing more explicit theoretical predictions, such as under what conditions should self-regulation act as a moderator and buffer stress effects, and why should it mediate self-regulatory resources?
Response:
We expanded the theoretical explanation for mediation (H4.3) by linking appraisal processes, regulatory depletion, and well-being outcomes. We also clarified the rationale for H1 by elaborating on class-level influences earlier in the Introduction. We expanded the logic of why both mechanisms may operate: mediation via regulatory depletion, and moderation via regulatory resources. This provides clearer justification for simultaneous testing.
“The present study addresses these gaps by investigating the role of self-regulation in the association between school stress and students’ school well-being in Grades 4, 6, and 8. Specifically, we test two complementary mechanisms: (1) mediation — whether self-regulation (or difficulties in self-regulation) helps explain how perceived school stress and classroom-level exposure to stressors translate into lower school well-being; and (2) moderation — whether stronger self-regulation attenuates the negative association of both perceived school stress and classroom-level stressor exposure with well-being.
In addition, we examine the direct associations of perceived school stress and classroom-level exposure to stressors with students’ school well-being. By simultaneously considering the school as a potential source of stress and as a critical setting for well-being promotion, the study aims to advance transactional stress theory and provide evidence-based insights for school policy and preventive interventions.
While perceived stress (the subjective appraisal of stressors) has been widely studied in relation to well-being, research suggests that classroom-level exposure to stressors may also be associated with students’ lower well-being, even when they do not explicitly perceive certain stressors as stressful (Slimmen et al., 2022). Perceived school stress reflects students’ subjective appraisal of how stressful the experienced school demands are, whereas exposure to school stressors captures the frequency of stressful events occurring in the classroom environment, irrespective of how stressful they were perceived to be. These constructs are related but not identical: students may be exposed to the same stressors but differ in how strongly they appraise them, and classrooms may differ systematically in the overall level of stressor exposure regardless of individual perceptions. Distinguishing these two aspects of stress allows us to model both the subjective experience of stress at the individual level and the objective or structural conditions of the classroom environment at the group level.
The distinction between reported exposure to stressors (frequency of their occurance) and their perceived stressfulness therefore reflects two theoretically meaningful aspects of the stress process: the environmental demands that students collectively encounter and the individual appraisals through which those demands are interpreted. Incorporating both levels into a multilevel model allows us to capture how classroom-level stressor exposure may contribute to differences in well-being. Therefore, this study examines the relationship between stress and well-being from two perspectives: perceived school stress and exposure to school stressors. To capture these dynamics, we employ hierarchical linear modeling (HLM) to differentiate between individual-level perceived stress and classroom-level exposure to stressors (see Figure 1).”
“To clarify the theoretical mechanism underlying our hypotheses, we situate self-regulation within two complementary frameworks: the transactional model of stress and coping (Lazarus & Folkman, 1984) and the contextual model of self-regulation (Murray et al., 2019). Drawing on the transactional model, we conceptualise stress as a process in which students’ cognitive appraisals of school demands determine the degree to which these demands are experienced as stressful, and these stress experiences, in turn, shape students’ emotional and behavioural responses. Within this framework, appraisals that result in higher levels of perceived stress are expected to be accompanied by more frequent negative emotions and dysregulated reactions, which in turn relate to lower well-being, providing a theoretical basis for our mediation hypothesis.
The contextual model further emphasises that self-regulation develops through ongoing interactions with environmental stressors and supports, and that chronic stress may deplete regulatory resources through cognitive overload and emotional strain. Because of this, students who have stronger regulatory skills are generally better able to manage stress, whereas students with weaker regulatory skills may be more affected by the same stressful experiences — which is the idea behind our moderation hypothesis. Together, these frameworks therefore justify examining self-regulation both as a process through which stress-related experiences may affect well-being and as a capacity that may modify the strength of this association.
When stress is frequent or intense enough to tax emotional and cognitive resources, it may reduce students’ regulatory capacity, which in turn affects their well-being — a pattern consistent with a mediation mechanism. At the same time, self-regulation also reflects a capacity to manage emotional arousal and maintain goal-directed behaviour; therefore, students with stronger self-regulation may be cope better with the negative effects of stress, whereas those with weaker self-regulation may be more affected, which aligns with a moderation mechanism. These two pathways are theoretically compatible rather than mutually exclusive, making it relevant to examine both within the same model.”
Reviewer 1 – Comment 7:
Classroom level mechanisms - While the authors justify measuring classroom-level stressor exposure, they provide limited theoretical explanation for why classroom-level factors would matter beyond individual perceptions. The social norms argument (line 74-77) is mentioned but underdeveloped. In addition, the manuscript cites Folkman's transactional stress model but doesn't fully develop how this framework informs the study design. The distinction between stressor exposure and perceived stress aligns with stress-appraisal theories, but this connection isn't made explicit.
Response:
We expanded the discussion of classroom-level influences, social norms, and environmental demands, and explicitly tied the distinction between exposure and appraisal to the transactional model of stress.
“However, not all students experience school stress in the same way. Students’ responses to stressors depend on how they interpret the situation and the resources available to manage it (Folkman, 1984). For instance, highly reactive individuals may perceive even minor stressors as overwhelming, increasing their risk of psychological distress and the need for structured support (Boyce & Ellis, 2005). More broadly, students’ responses to stressors vary based on their coping mechanisms, personality, and social expectations (Aldwin, 2007; Högberg, 2021). Those with high neuroticism, or maladaptive coping strategies, such as avoidance or emotional suppression, tend to experience stronger stress reactions. Additionally, older students and girls typically report higher stress, possibly due to greater academic expectations (Löfstedt et al., 2020).
According to the transactional model of stress (Lazarus & Folkman, 1984), stress arises from the interaction between environmental demands and the individual’s primary and secondary appraisals of those demands. Appraisal determines whether a situation is consciously experienced as stressful. However, research in neurobiology and psychophysiology shows that repeated or persistent exposure to demanding or conflictual environments can produce negative psychological and physiological consequences even when individuals do not report feeling stressed (McEwen, 1998, 2017). These effects occur through mechanisms such as low-intensity or automatic appraisals, cumulative allostatic load, gradual depletion of coping resources, and physiological activation that does not always reach conscious awareness. As a result, repeated exposure to classroom stressors (e.g., peer conflicts or high academic demands) may gradually contribute to increased strain, which in turn can undermine students’ well-being over time, even when they do not consciously appraise these situations as stressful.”
Reviewer 1 – Comment 8:
Inconsistent terminology: The manuscript alternates between "school stress," "school-related stress," and "school stress level." Standardize terminology.
Response:
Thank you for the comment. We revised terminology across the manuscript to ensure consistency.
Reviewer 1 – Comment 9:
Citation density: Some claims lack citations (e.g., line 169-170 on gender differences) while other sections are over-cited.
Response:
We added missing citations (“Högberg, 2021; Inchley et al., 2020”) to balance density of citations.
Reviewer 1 – Comment 10:
Construct clarity: The relationship between "self-regulation" and "coping" is unclear. Are these overlapping constructs or distinct?
Response:
We added an explicit conceptual distinction between self-regulation and coping.
“Self-regulation is distinct from coping: it reflects students’ general capacity to manage emotions and behaviour across situations, whereas coping refers to specific strategies used in response to a particular stressful event (e.g., problem-solving, avoidance, seeking support).”
Reviewer 1 – Comment 11:
No likelihood ratio test are reported.
Response:
The requested likelihood-based information has now been added to the table, including the log-likelihood values and the MLR scaling correction factors for all models.
Reviewer 1 – Comment 12:
The authors should add explicit statements and an explanation of the models. Please explain the difference between model 3-4 and model 6. Please add formal mediation statistics such as proportion mediated Sobel test.
Response:
The requested likelihood-based information has now been added to the table, including the log-likelihood values and the MLR scaling correction factors for all models.
(401-406) We clarified the conceptual and statistical differences between the regression models (Models 3–4) and the multilevel mediation model (Model 6): “Finally, a two-level path model was estimated to explicitly examine the mediating role of self-regulation in the stress–well-being link (H4.3), incorporating both direct and indirect effects, with sex (L1), grade, and class-level exposure to stressor (L2) as controls. Variables were pre-standardized, so coefficients were interpreted accordingly. Analyses were conducted in Mplus 8.11 (Muthén & Muthén, 2017) using the robust maximum likelihood estimator (MLR), with model fit assessed via AIC and sample-size adjusted BIC. Model 6 differs conceptually and statistically for Models 3–4 because it: simultaneously estimates the effect of perceived stress on self-regulation and the effect of self-regulation on school well-being; produces formal indirect (mediated) effects, including their standard errors and significance levels; uses the multilevel structure (L1 and L2) to specify mediation occurring within students, while controlling for class-level predictors. Thus, Models 3–4 show how coefficients change when predictors are added, whereas Model 6 formally tests mediation, decomposing the total effect of school stress into direct and indirect components.”
(464) In the description of the results, we have added:
“The Sobel test confirmed that the indirect effect of perceived school stress on school well-being via self-regulation was statistically significant (z = –7.28, p < .001), supporting the results obtained with the multilevel path model.

Reviewer 2 Report
Comments and Suggestions for Authors
See attached

Author Response
Reviewer 2– Comment 1:
How does school well-being is defined for the purposes of the study?
Response:
We added a full definitional subsection on school well-being.
“School well-being is a multidimensional construct that reflects how students experience and evaluate their life at school. Previous research distinguishes four broad components of school well-being: emotional (positive feelings at school), social (supportive peer and teacher relationships), academic (perceived competence and engagement), and physical or environmental aspects (feeling safe and comfortable at school; Hascher, 2012; Konu & Rimpelä, 2002). Rather than representing the absence of distress, school well-being emphasises positive functioning, such as enjoyment of learning, belonging, motivation, and supportive relationships with peers and teachers (Renshaw et al., 2015; Graham et al., 2017). School well-being is shaped by students’ overall psychological well-being, as their mental state affects how they experience and engage with the school environment (Morosanova et al., 2021).
High school well-being has been linked to stronger academic engagement and more adaptive social behaviour, and, according to some studies, to lower absenteeism (Cefai et al., 2021; López et al., 2022; Siebecke, 2024). Students with high school well-being are typically more involved and motivated at school, which is reflected in greater participation, higher academic achievement, and stronger peer and teacher relationships”
Reviewer 2– Comment 2:
The literature review section does not include studies on Primary education with an explicit focus on the relationship between self-RG and student well-being.
Response:
We added relevant primary-school literature and clarified developmental context.
“Strong self-regulation is associated with positive outcomes across multiple domains, including academic achievement, social competence, and mental health (Robson et al., 2020). Research consistently indicates that self-regulation is a significant predictor of psychological well-being in adolescents, with interventions targeting self-regulatory techniques showing moderate-to-large effects on mental health outcomes (van Genugten et al., 2017). Studies using self-regulation measures in school-aged children show that higher self-regulation is associated with greater emotional well-being, fewer internalizing and externalizing difficulties, and more positive relationships with peers and teachers (Fomina et al., 2020). Longitudinal work further indicates that self-regulation supports the development of both psychological and school-related well-being as children progress through late primary and early secondary school (Morosanova et al., 2021; Morosanova et al., 2023). These findings suggest that self-regulation already functions as an important correlate of well-being in primary school.”
Reviewer 2– Comment 3:
It doesn’t state the significance of the study, it doesn’t say anything as to why student well-being is worth exploring within the Polish primary education context.
Response:
We added a new paragraph situating the study within the Polish educational context and national data.
“Student well-being requires special attention in the Polish primary-school context, given rising evidence from national surveys indicating elevated levels of school-related stress and psychological difficulties among children and adolescents. For instance, the 2021/2022 Health Behaviour in School-aged Children (HBSC) survey revealed rising school pressure, particularly among girls, with Poland ranking poorly in adolescent well-being metrics across Europe (Inchley et al., 2024). The fact that over 40% of Polish students report frequent nervousness and low mood, often linked to school dissatisfaction and perceived high academic demands, underscores the urgency of targeted research. By examining how individual self-regulation capacities, perceived school stress and school stressors exposure jointly relate to student well-being, the present study provides actionable evidence for prevention efforts, educational policy, and school-based interventions to support mental health in Polish elementary education.”
Reviewer 2– Comment 4:
The results of this study (focusing on the relationship of self-regulation + student wellbeing in primary education) presented in the section are not at all discussed in the light of similar studies conducted in the same grade.
Response:
Thank you for this comment. We tried to address it adding a paragraph to the discussion.
“These findings are generally consistent with prior research conducted with primary school students, although direct comparisons remain limited. Studies using similar age groups have shown that higher self-regulation is associated with greater emotional well-being and more positive school functioning (Rodriguez et al., 2022; Valiente et al., 2020). Furthermore, studies examining school stress in upper primary grades have documented that students who report more intense or frequent stressors also tend to report lower school satisfaction and belonging (Hascher, 2012), broadly supporting the pattern identified in our results. Although few studies have tested mediation or mod-eration models in this age group, the direction and strength of the associations observed in the present study are consistent with these earlier findings.”
Reviewer 2– Comment 5:
Were your instruments translated and accommodated based on the context and age of your participants? You say nothing about it in the Methods section
Response:
We clarified that all instruments were validated Polish versions adapted for age, citing psychometric details.

Reviewer 3 Report
Comments and Suggestions for Authors
Review Report
The manuscript addresses a highly relevant topic in contemporary educational psychology: the relationship between school stress and school well-being in primary school students, incorporating the role of self-regulation both as a mediator and as a potential moderator, and explicitly distinguishing the individual level (L1) from the classroom level (L2) through hierarchical linear modeling. The use of recent, well-described instruments (SSWQ-PL-R, sSRS, SESQ) and a multilevel design with 702 Polish students in Grades 4, 6, and 8 provides solid empirical grounding for the study. Overall, the findings offer consistent evidence that self-regulation is positively associated with school well-being and partly mediates the association between perceived stress and well-being, while classroom-level exposure to stressors remains a relevant predictor of average well-being.
The strengths of the manuscript include: (a) an up-to-date theoretical framework; (b) a clear conceptual and analytical distinction between perceived school stress and classroom-level exposure to stressors, which makes it possible to capture both subjective and contextual components of stress; (c) a transparent, stepwise analytical strategy (Models 0–6) that allows readers to follow how covariates, predictors, and interactions are added; and (d) a discussion that, overall, acknowledges the correlational nature of the data and situates the findings within the international literature on school well-being and school stress. However, for the full potential of the study to be realized, the manuscript would benefit from some conceptual and statistical clarifications, as well as minor structural refinements. Below are specific suggestions, organized by section.
Title
The current title is informative but rather long and dense. You might consider shortening and streamlining it to improve readability and focus. For instance, the phrase “The Multilevel Analysis in Individual and Classroom Contexts” partly repeats information that is already conveyed in the Method section and in the abstract.
As an illustration (only as a suggestion), a more concise title could be along the lines of: “Self-Regulation as a Mediator and Moderator between School Stress and School Well-Being: A Multilevel Study”
or even: “Self-Regulation, School Stress, and School Well-Being: A Multilevel Analysis”
Of course, the exact wording is up to you, but a shorter and more focused title would likely make the manuscript more accessible to readers.
Abstract
The abstract is clear and well aligned with the content of the manuscript: it defines the main focus (school stress, school well-being, and self-regulation), identifies the primary analytic technique (HLM), and anticipates the key findings (associations at both individual and classroom level; mediation but no moderation by self-regulation). Nonetheless, it could be strengthened by greater methodological precision and more cautious causal language.
- Structure and content
- Explicitly state the study design. For example:
“In a cross-sectional study using hierarchical linear modeling…” - Specify the country and approximate age range in addition to the grade levels, for instance:
“702 Polish primary school students (Grades 4, 6, and 8; approx. ages 10–15)”. - Clarify in a single sentence that two indicators of school stress are examined:
“individual-level perceived school stress and classroom-level exposure to school stressors.”
- Explicitly state the study design. For example:
- Causal language
- Consider revising expressions such as “mediated the relationship” and “impact of stress on well-being”, which may imply causality in a cross-sectional design. More cautious alternatives could be:
- “partly accounted for the association between perceived stress and well-being”
- “helps explain the link between stress and well-being.”
- Similarly, “buffers the effects of stress” also reads as causal. A more prudent phrasing might be:
“attenuate the association between stress and well-being.”
- Consider revising expressions such as “mediated the relationship” and “impact of stress on well-being”, which may imply causality in a cross-sectional design. More cautious alternatives could be:
Introduction
The introduction is well documented and provides a clear overview of the constructs of school stress, school well-being, and self-regulation, supported by recent and relevant references. Even so, the section could benefit from a more compact and structured presentation of the aims and hypotheses.
- It would be helpful to reinforce the conceptual definitions of the core variables (particularly school stress, but also school well-being and self-regulation) before moving on to their associations with other factors. This would help to anchor the subsequent multilevel model more firmly.
- I recommend presenting the hypotheses in a numbered list (H1, H2, H3, H4, …) rather than embedding them in dense paragraphs. This would facilitate alignment with the study aims and allow for a clearer comparison when the hypotheses are revisited in the Results and Discussion sections.
- I suggest relocating the conceptual model to the end of the Introduction, immediately after the hypotheses, rather than in the Data Analysis section. In that position, the diagram can:
- clearly illustrate the assumed relationships among school stress, self-regulation, and school well-being,
- visually differentiate between mediation and moderation expectations, and
- strengthen the transition from the theoretical framework to the analytical strategy.
- It would be beneficial to state more explicitly the specific gap that this study addresses (i.e., the scarcity of multilevel models integrating individual-level stress, classroom-level stress, self-regulation, and school well-being in primary school populations).
- The manuscript should avoid alternating between the terms “gender” and “sex” without distinction. Given that the study relies on self-report data rather than biological markers, it would be preferable to use the term “gender”
Methods
The methodology is generally rigorous and clearly presented. Nonetheless, several aspects would benefit from further precision:
- In the description of the sample and context, it would be useful to provide more detail about the classroom-group structure (e.g., average class size, distribution by grade) and the recruitment procedure (e.g., convenience sampling, participating schools, inclusion criteria).
- As the participants are minors, the procedures for obtaining informed consent should be described more explicitly, including parental consent and students’ assent, as well as the guarantees of confidentiality and data protection.
- In the instruments section, it would be helpful to emphasize that the self-regulation measure primarily reflects self-regulatory difficulties (rather than the presence of adaptive self-regulatory strategies). This clarification can prevent subsequent interpretations from presenting self-regulation in an exclusively “protective” light.
- The Data Analysis section describes Models 0–6 clearly, but it could be strengthened by:
- explicitly stating the key assumptions underlying multilevel modeling (e.g., ICC values and the justification for using HLM instead of single-level regression models),
- briefly justifying the use of second-order theta scores instead of directly using items or first-order factor scores, and
- explicitly noting that no between-group invariance analyses were conducted (which might be relevant when comparing across grades or gender).
Results
The results are presented in an orderly manner, following the sequence of multilevel models, which makes it easy to understand how predictors are added and model fit improves.
- It might be useful to include (perhaps as supplementary material) a table with descriptive statistics (M, SD) and bivariate correlations among the main study variables to give readers a preliminary overview of the associations.
- In presenting the models, the reporting of coefficients and significance levels is appreciated, but the interpretation of effect sizes could be further developed (for example, by offering some indication of the relative magnitude of changes in well-being associated with one standard deviation change in stress or self-regulation).
- The moderation test (Stress × Self-regulation interaction) is correctly implemented, but it would be helpful to state briefly in the Results section that the interaction term was not significant and that, therefore, the “buffering” hypothesis is not empirically supported in this dataset.
- Table 1 should be formatted so that it fits on a single page.
Discussion
The Discussion section is well structured and provides a nuanced interpretation of the findings, but it could be strengthened in several ways:
- It would be worthwhile to reinforce the connection with the transactional model of stress (Lazarus & Folkman, 1986), explicitly discussing how the results can be interpreted in terms of demands, resources (self-regulation), and well-being outcomes.
- Given that self-regulation is assessed largely as difficulty or deficit, I would suggest tempering the idea of a “protective factor” and instead referring to self-regulation as a self-regulatory resource, whose presence or absence is associated with higher or lower levels of well-being.
- When interpreting the findings in terms of the “impact” or “buffering” of stress, it would be useful to reiterate explicitly the cross-sectional and correlational nature of the data, avoiding formulations that might be read as strong causal claims—particularly regarding moderation, which was not supported by the analyses.
- The discussion of school-based implications is appropriate and relevant; it could be enriched with more concrete examples of classroom practices or school-wide programs aimed at promoting self-regulation and reducing stressors (e.g., organizational adjustments, classroom climate initiatives, teacher training).
Conclusions
The Conclusions section appropriately summarizes the main findings but could be refined as follows:
- Clearly distinguish the summary of empirical results (empirical level) from broader implications (conceptual and applied levels).
- Maintain a strong focus on the scientific contribution of the study (a multilevel model integrating stress, self-regulation, and school well-being) and avoid overly programmatic recommendations unless they are explicitly linked to the data obtained.
- Close with a sentence that underscores the contribution to knowledge on school well-being and self-regulation in primary school students, while also highlighting the need for future research (e.g., longitudinal designs or experimental interventions).
Comment on structure and section headings
To improve readability and alignment with common conventions in empirical papers, I would recommend slightly restructuring and simplifying the section headings. In several cases, the current titles are very long and include information that could be better placed in the text rather than in the heading itself. For example:
- Introduction
I suggest keeping the main heading as 1. Introduction and using concise subheadings, such as: - 1.1 School Stress
- 1.2 School Well-Being
- 1.3 Self-Regulation
- 1.4 The Present Study (here you could briefly state the gap, list the hypotheses in numbered form, and show the conceptual model).
- Method
It might be clearer to organize this section as follows: - 2.1 Design and Participants – explicitly state the study design (e.g., cross-sectional), and add the age of participants (M, SD, range).
- 2.2 Procedure – describe the data collection procedure and indicate when the study was conducted (school year, months), as well as consent procedures.
- 2.3 Measures – I would keep the heading short (without “variables”) and maintain a clear numbering of all instruments. For example, if the first subsections already cover school well-being and self-regulation (2.3.1–2.3.3), then:
- 2.3.4 School Stressors Exposure
- 2.3.5 School Stress Level
- 2.3.6 Demographic Covariates
- 2.4 Statistical Analysis – this shorter title may be preferable to the longer “Analytical Approach and Model Descriptions”; the detailed description of Models 0–6 can remain in the text.
- Results
The heading “Results of Multilevel Linear Regression Analyses” could be simplified to 3. Results, as the type of analysis is already clear from the Method section. - Discussion and conclusions
For a cleaner structure, you might consider: - 4. Discussion
- 4.1 Theoretical and Practical Implications
- 4.2 Limitations and Future Research Directions
- 5. Conclusions
This reorganization would make the manuscript easier to navigate, reduce redundancy in the headings, and highlight more clearly the logic from theory → method → results → implications.
Overall Assessment
This is a well-grounded manuscript that contributes to understanding how school stress—both perceived and contextual—is related to students’ well-being, and how self-regulation may play a role in this association. With the suggested revisions—mainly aimed at refining the interpretation, clarifying certain technical aspects of the multilevel modeling, and polishing the wording at key points—the manuscript has the potential to become a relevant reference in the field of school mental health and research on well-being in educational settings.

Author Response
Reviewer 3 – Comment 1:
It would be helpful to reinforce the conceptual definitions of the core variables (particularly school stress, but also school well-being and self-regulation) before moving on to their associations with other factors. This would help to anchor the subsequent multilevel model more firmly.
Response:
We strengthened conceptual definitions at the beginning of the Introduction.
“School well-being is a multidimensional construct that reflects how students experience and evaluate their life at school. Previous research distinguishes four broad components of school well-being: emotional (positive feelings at school), social (supportive peer and teacher relationships), academic (perceived competence and engagement), and physical or environmental aspects (feeling safe and comfortable at school; Hascher, 2012; Konu & Rimpelä, 2002). Rather than representing the absence of distress, school well-being emphasises positive functioning, such as enjoyment of learning, belonging, motivation, and supportive relationships with peers and teachers (Renshaw et al., 2015; Graham et al., 2017). School well-being is shaped by students’ overall psychological well-being, as their mental state affects how they experience and engage with the school environment (Morosanova et al., 2021).”
“Students respond to school stressors with varying degrees of self-regulation, which involves managing thoughts and emotions in ways that support goal-directed behavior (Murray et al., 2019). Self-regulation involves processes such as executive functions, inhibitory control, and emotion regulation, all of which work together to support adaptive functioning. Self-regulation is distinct from coping: it reflects students’ general capacity to manage emotions and behaviour across situations, whereas coping refers to specific strategies used in response to a particular stressful event (e.g., problem-solving, avoidance, seeking support).
“Strong self-regulation is associated with positive outcomes across multiple domains, including academic achievement, social competence, and mental health (Robson et al., 2020). Research consistently indicates that self-regulation is a significant predictor of psychological well-being in adolescents, with interventions targeting self-regulatory techniques showing moderate-to-large effects on mental health outcomes (van Genugten et al., 2017). Studies using self-regulation measures in school-aged children show that higher self-regulation is associated with greater emotional well-being, fewer internalizing and externalizing difficulties, and more positive relationships with peers and teachers (Fomina et al., 2020). Longitudinal work further indicates that self-regulation supports the development of both psychological and school-related well-being as children progress through late primary and early secondary school (Morosanova et al., 2021; Morosanova et al., 2023). These findings suggest that self-regulation already functions as an important correlate of well-being in primary school.”
Reviewer 3 – Comment 2:
It would be beneficial to state more explicitly the specific gap that this study addresses (i.e., the scarcity of multilevel models integrating individual-level stress, classroom-level stress, self-regulation, and school well-being in primary school populations).
Response:
We added an explicit gap statement highlighting the novelty of integrating multilevel stress, self-regulation, and well-being in primary school.
“Although many studies have examined links between self-regulation, stress, and well-being, the specific role of self-regulation as a mediator or moderator in these associations remains insufficiently understood, especially in primary school populations. Existing research with adolescents and adults suggests that self-regulation may mediate the association between stress and well-being (Rollins & Crandall, 2021; Yazıcı-Kabadayı & Öztemel, 2024) and may also moderate it by buffering or amplifying stress effects (Kadzikowska-Wrzosek, 2012; Rufino et al., 2022). However, these studies differ widely in their conceptualisations of well-being and often operationalise it as the absence of distress or psychosomatic symptoms, providing a limited view of positive functioning. From a positive development perspective, well-being encompasses not only low ill-being but also growth, engagement, and belonging (Ruini et al., 2003). Research examining how self-regulation relates to these broader dimensions of school well-being is scarce, and even fewer studies assess both mediation and moderation within the same model.
Moreover, multilevel studies that simultaneously consider perceived school stress and exposure to school stressors in primary school students are particularly rare, despite the fact that classroom dynamics are known to play a significant role in children's daily experiences at school. Very little is known about how these multilevel stress processes operate in students. Clarifying these mechanisms is therefore essential for advancing theoretical understanding and informing interventions tailored to primary school learners.”
Reviewer 3 – Comment 3:
As an illustration (only as a suggestion), a more concise title could be along the lines of: “Self-Regulation as a Mediator and Moderator between School Stress and School Well-Being: A Multilevel Study
Response:
We thank the reviewer for the suggestion; we changed the title of the study as suggested.
Reviewer 3 – Comment 4:
Abstract: Nonetheless, it could be strengthened by greater methodological precision and more cautious causal language.
Structure and content - Explicitly state the study design. For example:
“In a cross-sectional study using hierarchical linear modeling…”
Specify the country and approximate age range in addition to the grade levels, for instance: “702 Polish primary school students (Grades 4, 6, and 8; approx. ages 10–15)”. Clarify in a single sentence that two indicators of school stress are examined:
“individual-level perceived school stress and classroom-level exposure to school stressors.” Causal language - consider revising expressions such as “mediated the relationship” and “impact of stress on well-being”, which may imply causality in a cross-sectional design. More cautious alternatives could be:
“partly accounted for the association between perceived stress and well-being”
“helps explain the link between stress and well-being.”
Similarly, “buffers the effects of stress” also reads as causal. A more prudent phrasing might be:“attenuate the association between stress and well-being.”
Response:
We thank the reviewer for the suggestion; we improved clarity in the abstract, results and discussion to avoid causal language.
“This study examines the relationship between school stress and school well-being, focusing on the mediating and moderating role of self-regulation. This cross-sectional study uses hierarchical linear modeling to assess how two aspects of school stress - perceived school stress at the individual level (students’ subjective appraisal of how stressful specific school demands are) and classroom stressor exposure at the group level (the aggregated frequency of stressful events occurring in each classroom) are linked to student school well-being. The sample included 702 Polish primary school students (Grades 4, 6, and 8, approx. ages 10–15). Results indicate that while higher perceived school stress is associated with lower well-being, classroom-level stressor exposure also contributes to variations in student well-being. Self-regulation was positively associated with school well-being and partly accounted for the association between perceived stress and well-being. However, no significant moderating effect of self-regulation was found, suggesting that while self-regulation helps explain the link between stress and well-being., it does not necessarily attenuate the association between stress and well-being. These findings highlight the importance of both individual self-regulation skills and structural interventions aimed at reducing classroom stressors to promote student well-being.”
Reviewer 3 – Comment 5:
I suggest relocating the conceptual model to the end of the Introduction, immediately after the hypotheses, rather than in the Data Analysis section. In that position, the diagram can:
clearly illustrate the assumed relationships among school stress, self-regulation, and school well-being, visually differentiate between mediation and moderation expectations, and strengthen the transition from the theoretical framework to the analytical strategy.
Response:
We relocated the conceptual model to the Introduction.
Reviewer 3 – Comment 6:
In the description of the sample and context, it would be useful to provide more detail about the classroom-group structure (e.g., average class size, distribution by grade) and the recruitment procedure (e.g., convenience sampling, participating schools, inclusion criteria).
Response:
We rewrote the participants section.
“The study was conducted in public primary schools located in a medium-sized city in central Poland. In this context, students are educated in stable classroom groups that remain together across grades, with teachers assigned to specific class divisions. In our sample, class sizes ranged from 5 to 25 students (M ≈ 13.3). The final dataset included 52 classroom groups: 21 in Grade 4, 21 in Grade 6, and 10 in Grade 8.
Schools were recruited through cooperation with the local education authority, which distributed study invitations to all public primary schools in the municipality. Participation was voluntary and based on convenience sampling, as schools self-selected into the study. Within participating schools, all students in the target grades were invited to participate, provided that informed parental consent was obtained. No exclusion criteria were applied other than the inability to complete the online questionnaire during the scheduled data collection session.
The sample included 702 Polish adolescents (49.3% girls) from Grades 4 (45.7%), 6 (35.9%), and 8 (18.4%), typically aged 10–15. Participants were drawn from all public primary schools in a medium-sized city in central Poland, offering a comprehensive view of the student population in that area. In Poland, education begins with a one-year pre-primary program at age six, followed by eight years of primary school (Grades 1–8). After primary school, students take an external exam, which is crucial for secondary school admission.”
Reviewer 3 – Comment 7:
As the participants are minors, the procedures for obtaining informed consent should be described more explicitly, including parental consent and students’ assent, as well as the guarantees of confidentiality and data protection.
Response:
We added this clarification to the description of the tool.
“This predictor variable was measured using the 12-item Polish adaptation of the Self-Regulation Scale (sSRS; (RodzeÅ„ & Gajda, 2024), developed originally by Novak and Clayton (2001). The scale comprises three subscales: managing anger and frustration (e.g., "I have difficulty controlling my temper"), goal-setting and planning (e.g., "Once I have a goal, I make a plan how to reach it"), and impulse control in goal-directed situations (e.g., "I get very fidgety after a few minutes if I am supposed to sit still"). Items are rated on a 4-point Likert scale (1 = "never true" to 4 = "always true"). A higher-order CFA showed acceptable fit (RMSEA = 0.068), and reliability for the total score was αord = 0.82.
Because direct measurement of adaptive self-regulatory strategies can be difficult in primary-school children, the instrument primarily captures self-regulatory difficulties (dysregulation), such as problems with attention, emotional control, or behavioural inhibition. The statements for the emotional and behavioral dimensions are scored inversely. Thus, higher scores reflect higher self-regulation.”
Reviewer 3 – Comment 8:
In the instruments section, it would be helpful to emphasize that the self-regulation measure primarily reflects self-regulatory difficulties (rather than the presence of adaptive self-regulatory strategies). This clarification can prevent subsequent interpretations from presenting self-regulation in an exclusively “protective” light.
Response:
We added this clarification to the description of the tool.
“This predictor variable was measured using the 12-item Polish adaptation of the Self-Regulation Scale (sSRS; (RodzeÅ„ & Gajda, 2024), developed originally by Novak and Clayton (2001). The scale comprises three subscales: managing anger and frustration (e.g., "I have difficulty controlling my temper"), goal-setting and planning (e.g., "Once I have a goal, I make a plan how to reach it"), and impulse control in goal-directed situations (e.g., "I get very fidgety after a few minutes if I am supposed to sit still"). Items are rated on a 4-point Likert scale (1 = "never true" to 4 = "always true"). A higher-order CFA showed acceptable fit (RMSEA = 0.068), and reliability for the total score was αord = 0.82.
Because direct measurement of adaptive self-regulatory strategies can be difficult in primary-school children, the instrument primarily captures self-regulatory difficulties (dysregulation), such as problems with attention, emotional control, or behavioural inhibition. The statements for the emotional and behavioral dimensions are scored inversely. Thus, higher scores reflect higher self-regulation.”
Reviewer 3 – Comment 9:
Table 1 should be formatted so that it fits on a single page.
Response:
In the revised version of the manuscript, we have reformatted Table 1 to ensure that it fits on a single page.
Reviewer 3 – Comment 10:
It would be worthwhile to reinforce the connection with the transactional model of stress (Lazarus & Folkman, 1986), explicitly discussing how the results can be interpreted in terms of demands, resources (self-regulation), and well-being outcomes.
Response:
Thank you for this suggestion. We integrated the transactional model of stress in the Introduction, where we describe how students’ appraisals of school demands contribute to their experience of stress. We decided to highlight the connection with the transactional model of stress in the discussion by adding another paragraph.
“At the individual level, the observed mediation pattern is compatible with the transactional model of stress (Lazarus & Folkman, 1984), which describes stress as arising from the appraisal of environmental demands relative to personal resources. In our findings, students who appraised school demands as more stressful also tended to report lower self-regulation, aligning with the notion that heightened stress experiences may coincide with reduced regulatory capacity. Lower self-regulation, in turn, was associated with lower school well-being. This suggests that appraisal-based stress is linked to well-being partly through students’ self-regulatory resources.”
Reviewer 3 – Comment 11:
Given that self-regulation is assessed largely as difficulty or deficit, I would suggest tempering the idea of a “protective factor” and instead referring to self-regulation as a self-regulatory resource, whose presence or absence is associated with higher or lower levels of well-being.
Response:
We revised terminology throughout the manuscript to avoid describing self-regulation as a protective factor.
Reviewer 3 – Comment 12:
When interpreting the findings in terms of the “impact” or “buffering” of stress, it would be useful to reiterate explicitly the cross-sectional and correlational nature of the data, avoiding formulations that might be read as strong causal claims—particularly regarding moderation, which was not supported by the analyses.
Response:
We thank the reviewer for highlighting the need to avoid causal language. In response, we replaced the term “buffering” with the more neutral phrasing “attenuates the association” throughout the manuscript. We also added an explicit reminder in the Discussion that the data are cross-sectional and correlational, and that moderation was not supported by the analyses. These revisions clarify the interpretive limits of the study and align the wording with the statistical evidence.
Revision in the manuscript (exact changed/added fragments):
Causal wording replaced:
“…we assess whether self-regulation attenuates the association between stress and school well-being…”
Moderation caveat added in the Discussion:
“…the interaction effect was not statistically significant, suggesting that self-regulation did not attenuate the association between perceived stress and well-being in this sample.”
Also, in limitation section cross-sectional caution is stated:
“…it is important to note that these associations are based on cross-sectional and correlational data, and therefore do not permit strong causal interpretations.”
Reviewer 3 – Comment 13:
CONCLUSION: Clearly distinguish the summary of empirical results (empirical level) from broader implications (conceptual and applied levels).
Maintain a strong focus on the scientific contribution of the study (a multilevel model integrating stress, self-regulation, and school well-being) and avoid overly programmatic recommendations unless they are explicitly linked to the data obtained.
Close with a sentence that underscores the contribution to knowledge on school well-being and self-regulation in primary school students, while also highlighting the need for future research (e.g., longitudinal designs or experimental interventions).
Response:
We thank the reviewer for this valuable suggestion. We revised the Conclusion to explicitly separate the empirical summary from the conceptual and applied implications. We also strengthened the final section to highlight the scientific contribution of the multilevel model and added a closing statement on the need for future longitudinal or experimental research. These revisions improve the clarity, structure, and focus of the Conclusion.
Reviewer 3 – Comment 14:
Restructuring and simplifying the section headings
Introduction
I suggest keeping the main heading as 1. Introduction and using concise subheadings, such as:
1.1 School Stress
1.2 School Well-Being
1.3 Self-Regulation
1.4 The Present Study (here you could briefly state the gap, list the hypotheses in numbered form, and show the conceptual model).
Method
It might be clearer to organize this section as follows:
2.1 Design and Participants – explicitly state the study design (e.g., cross-sectional), and add the age of participants (M, SD, range).
2.2 Procedure – describe the data collection procedure and indicate when the study was conducted (school year, months), as well as consent procedures.
2.3 Measures – I would keep the heading short (without “variables”) and maintain a clear numbering of all instruments. For example, if the first subsections already cover school well-being and self-regulation (2.3.1–2.3.3), then:
2.3.4 School Stressors Exposure
2.3.5 School Stress Level
2.3.6 Demographic Covariates
2.4 Statistical Analysis – this shorter title may be preferable to the longer “Analytical Approach and Model Descriptions”; the detailed description of Models 0–6 can remain in the text.
Results
The heading “Results of Multilevel Linear Regression Analyses” could be simplified to 3. Results, as the type of analysis is already clear from the Method section.
Discussion and conclusions
For a cleaner structure, you might consider:
- Discussion
4.1 Theoretical and Practical Implications
4.2 Limitations and Future Research Directions
- Conclusions
Response:
We simplified and shortened titles as suggested in the following sections: introduction, methods, and results.
Reviewer 3 – Comment 11:
The Data Analysis section describes Models 0–6 clearly, but it could be strengthened by: explicitly stating the key assumptions underlying multilevel modeling (e.g., ICC values and the justification for using HLM instead of single-level regression models),
Briefly justifying the use of second-order theta scores instead of directly using items or first-order factor scores, and explicitly noting that no between-group invariance analyses were conducted (which might be relevant when comparing across grades or gender).
Response:
The table was added as a supplement.
Reviewer 3 – Comment 11:
In presenting the models, the reporting of coefficients and significance levels is appreciated, but the interpretation of effect sizes could be further developed (for example, by offering some indication of the relative magnitude of changes in well-being associated with one standard deviation change in stress or self-regulation).
Response:
(458) Added:
“After rejecting the hypothesis regarding the interaction between stress level and self-regulation, we return to Model 4. Focusing on the main explanatory variables, we can conclude that, at the individual level, an increase in perceived stress is associated with a decrease in school well-being of approximately one-tenth of a standard deviation. In contrast, higher self-regulation predicts an increase in well-being of more than one-third of a standard deviation. At the group (classroom) level, a one–standard deviation increase in exposure to stressors is linked to a reduction in average well-being of roughly one-seventh of a standard deviation.”
Reviewer 3 – Comment 11:
The moderation test (Stress × Self-regulation interaction) is correctly implemented, but it would be helpful to state briefly in the Results section that the interaction term was not significant and that, therefore, the “buffering” hypothesis is not empirically supported in this dataset.
Response:
(454) The description of the results includes the following passage:
“Model 5 tested an interaction term (stress × self-regulation) to examine whether self-regulation moderates the relationship between stress and well-being. However, no significant moderating effect was detected, and fit indices (AIC, BIC) indicated no improvement over Model 4, suggesting that self-regulation does not function as a moderator in this context, providing no empirical support for H4.2, which was therefore rejected.”
In conjunction with the addition in (458), this is a clear interpretative clue for the reader.

Round 2
Reviewer 1 Report
Comments and Suggestions for Authors
Thank you for submitting the revised version of your manuscript. I appreciate the careful attention you have given to the previous comments and the improvements made throughout the paper. The revisions have strengthened the manuscript, and I now find it suitable for publication.
I have only one minor suggestion: please begin a new line on p. 4, line 181, within the section describing the present study.
Author Response
Reviewer 1
Comment: Please begin a new line on p. 4, line 181, within the section describing the present study.
Response: The requested line break has been introduced in the specified section.
Reviewer 2 Report
Comments and Suggestions for Authors
Well done on your work
Author Response
Reviewer 2
Comment: Well done on your work.
Response: We thank the Reviewer for this positive evaluation.
Reviewer 3 Report
Comments and Suggestions for Authors
The revised version of the manuscript shows a clear and substantial improvement. The authors have adequately addressed the reviewers’ comments, and in my view the paper is now suitable for publication in EJIHPE.
I only have a few minor observations:
(1) In the abstract there is a small typo in the phrase “between stress and well-being.,“ — please correct the punctuation (remove the comma after the period).
(2) From a layout perspective, you might consider slightly condensing or synthesising one of the paragraphs so that Figure 1 can be placed on page 11, avoiding the large blank space that currently appears there.
(3) In the sentence “By integrating individual- and classroom-level …”, an unnecessary hyphen has been introduced after “individual”.
Author Response
Reviewer 3
Comment: Correct the punctuation in the abstract: “between stress and well-being.,“.
Response: The punctuation has been corrected and the comma removed.
Comment: Consider condensing or synthesising one paragraph to ensure that Figure 1 appears on page 11 without a large blank space.
Response: We edited and slightly adjusted the layout to make Figure 1 appear on page 11.
Comment: Remove the unnecessary hyphen after “individual” in the sentence “By integrating individual- and classroom-level…”.
Response: The extraneous hyphen has been removed.